# Chicken Juice Enhances *C. jejuni* NCTC 11168 Biofilm Formation with Distinct Morphological Features and Altered Protein Expression

**DOI:** 10.3390/foods13121828

**Published:** 2024-06-11

**Authors:** Kidon Sung, Miseon Park, Jungwhan Chon, Ohgew Kweon, Angel Paredes, Saeed A. Khan

**Affiliations:** 1Division of Microbiology, National Center for Toxicological Research, U.S. Food and Drug Administration, Jefferson, AR 72079, USA; miseon.park@fda.hhs.gov (M.P.); oh-gew.kweon@fda.hhs.gov (O.K.); saeed.khan@fda.hhs.gov (S.A.K.); 2Department of Companion Animal Health, Inje University, Gimhae 50834, Republic of Korea; alvarmar@naver.com; 3Office of Scientific Coordination, National Center for Toxicological Research, U.S. Food and Drug Administration, Jefferson, AR 72079, USA; angel.paredes@fda.hhs.gov

**Keywords:** chicken juice, *C. jejuni*, biofilm, quantitative proteome

## Abstract

*Campylobacter jejuni* is the foodborne pathogen causing most gastrointestinal infections. Understanding its ability to form biofilms is crucial for devising effective control strategies in food processing environments. In this study, we investigated the growth dynamics and biofilm formation of *C. jejuni* NCTC 11168 in various culture media, including chicken juice (CJ), brain heart infusion (BHI), and Mueller Hinton (MH) broth. Our results demonstrated that *C. jejuni* exhibited a higher growth rate and enhanced biofilm formation in CJ and in 1:1 mixtures of CJ with BHI or MH broth compared to these measures in BHI or MH broth alone. Electron microscopy unveiled distinct morphological attributes of late-stage biofilm cells in CJ, including the presence of elongated spiral-shaped cells, thinner stretched structures compared to regular cells, and extended thread-like structures within the biofilms. Proteomic analysis identified significant alterations in protein expression profiles in *C. jejuni* biofilms, with a predominance of downregulated proteins associated with vital functions like metabolism, energy production, and amino acid and protein biosynthesis. Additionally, a significant proportion of proteins linked to biofilm formation, virulence, and iron uptake were suppressed. This shift toward a predominantly coccoid morphology echoed the reduced energy demands of these biofilm communities. Our study unlocks valuable insights into *C. jejuni*’s biofilm in CJ, demonstrating its adaptation and survival.

## 1. Introduction

*Campylobacter jejuni* is a prominent contributor to gastrointestinal foodborne illness worldwide [1]. In 2020, the average reported incidence of *Campylobacter* infection declined in the United States and most European countries compared to 2014–2019 [2]. However, the Czech Republic had the highest worldwide incidence that year (215 per 100,000 population). This was followed by Australia (146.8 per 100,000 in 2016) and New Zealand (126.1 per 100,000 in 2019). Populations most susceptible to *C. jejuni* infection include young children, elderly individuals, and immunocompromised patients [3]. Diarrhea, stomach pain, nausea, vomiting, fever, and headache are the most common clinical symptoms of campylobacteriosis [4]. However, some patients can develop severe complications such as Guillain–Barré syndrome, Miller Fisher syndrome, reactive arthritis, irritable bowel syndrome, celiac disease, and other related diseases [5]. The primary mode of human infection with *C. jejuni* is believed to occur through ingesting undercooked poultry and raw milk [6,7]. Further, handling and preparation of contaminated poultry can be associated with the cross-contamination of other food items, thereby increasing the risk of indirect *C. jejuni* infections [8].

Biofilms are dynamic communities of cells that aggregate to form microcolonies [9]. They generate extracellular polymeric substances (EPSs), consisting of exopolysaccharides, eDNA, proteins, lipids, and a range of other biomolecules. Biofilms eventually disperse from the surface, enabling the initiation of new biofilm formation cycles at different locations [10]. The formation and behavior of a biofilm are influenced by interactions between bacteria, between bacteria and EPSs, as well as between bacteria and the surface.

In comparison to planktonic cultures, biofilm communities exhibit disparities compared to planktonic cultures, showing not only variations in metabolic activity but also distinct phenotypes at different stages of development [10]. Also, compared to their planktonic counterparts, bacterial cells that exist in the biofilm state exhibit significantly higher resilience to antibacterial agents, inadequate nutrient levels, and host immune responses [11]. These characteristics are particularly significant in the food industry. During the processing of products, biofilms that form on food plant surfaces frequently contribute to contamination during the processing of products. The cells within biofilms exhibit distinct gene expression patterns compared to those of their planktonic cell counterparts. However, the findings from bacterial transcriptomic analysis do not always align with the identified proteins and their functional activities. The proteomic approach provides valuable insights into the presence of functional molecules. Recent advances in quantitative proteomics have made it possible to better characterize the proteomes in a wide range of bacteria [12,13].

*C. jejuni* has been detected in biofilms within the watering supplies and plumbing networks of animal processing plants and livestock facilities [7]. It not only forms monospecies biofilms, but can also integrate itself into biofilms that already exist as the secondary colonizer [14]. The capacity of *C. jejuni* to form biofilms is believed to serve as a protective mechanism, enabling these biofilms to withstand cleaning and disinfection procedures and supporting their long-term survival in food processing facilities [15]. The exudate from frozen raw meat, a liquid rich in proteins, carbohydrates, and other nutrients released from chicken meat, can contaminate surfaces in food processing settings, thereby facilitating the dissemination of foodborne pathogenic bacteria [16].

To replicate nutrient conditions during poultry processing, chicken juice (CJ) has been utilized as a relevant food-based model [17,18]. CJ is the inherent liquid discharged from commercially bought frozen whole chicken once it has undergone thawing. A previous study indicated that CJ has the potential to trigger transcriptional responses in *C. jejuni* [19]. Another report revealed that CJ, compared to brucella broth, has the capacity to enhance cell survival and facilitate increased biofilm production in *C. jejuni* [18]. This effect is attributed to CJ forming a conditioning layer on various surfaces, providing a robust foundation for the attachment and development of *C. jejuni* biofilms. Consequently, CJ’s presence in industrial food environments can complicate the prevention of food contamination with *C. jejuni*. To fully understand the impact of CJ on *C. jejuni* biofilm production, we investigated its influence on growth and biofilm formation and carried out a label-free quantitative proteome analysis using the advanced Q-Exactive HF-X Orbitrap mass spectrometer.

## 2. Materials and Methods

### 2.1. Bacterial Strains, Media, and Culture Conditions

The *C. jejuni* National Collection of Type Cultures (NCTC) strain 11168 was obtained from Kelli L. Hiett at the Agricultural Research Service, U.S. Department of Agriculture, Athens, GA. For the preparation of precultures, cells were streaked from −80 °C vials onto Mueller Hinton (MH) agar plates supplemented with 5% lysed sheep blood (Hardy Diagnostics, Santa Maria, CA, USA) and incubated in the DG250 Microaerophilic Workstation (Don Whitley Scientific, Shipley, UK) at 37 °C for 24 h. Afterward, the resulting single colony was resuspended in MH broth and cultured at 37 °C for 24 h.

### 2.2. Measurement of Growth

Frozen whole chickens were obtained from various supermarkets in Little Rock, Arkansas [20]. Chickens were left to thaw overnight in a refrigerated environment. Exudate from the thawed chickens was collected, filtered through sterile nylon filter mesh (100 µm, Thermo Fisher Scientific, Waltham, MA, USA), and underwent centrifugation to remove debris. Subsequently, the exudate was sterilized using a 0.2 μm Rapid-Flow sterile disposable bottle top filter with polyethersulfone (Thermo Fisher Scientific), centrifuged at 20,817× *g* for 1 h at 4 °C, and the resulting CJ was collected. Bacterial contamination was confirmed by inoculating 100 µL of CJ onto MH agar supplemented with 5% lysed sheep blood, which was incubated for 48 h at 37 °C. CJ was then diluted in different ratios, e.g., 1:1 or 1:9, with brain heart infusion (BHI) broth (Hardy Diagnostics) or MH broth. The *C. jejuni* cells were subjected to centrifugation at 20,817× *g* for 5 min at 4 °C. Next, the supernatant was discarded, and the pellets were rinsed thrice using phosphate-buffered saline (PBS) (MilliporeSigma, Burlington, MA, USA). The bacterial suspension of *C. jejuni* 11,168 was subsequently calibrated to attain an optical density (OD) of 0.09 at 600 nm in their respective growth media using a SmartSpec Plus UV/VIS Spectrophotometer (Bio-Rad, Hercules, CA, USA) and added to each well of 96-well plates (Corning Inc., Corning, NY, USA). Media without the *C. jejuni* cells were included as negative controls. To prevent evaporation, 200 µL of PBS was added to each peripheral well, and the plates were securely sealed with parafilm. Each plate was inserted in a Synergy HT Microplate Reader (BioTek Instruments, Winooski, VT, USA), which was located within the DG250 Microaerophilic Workstation. Bacteria were subjected to continuous shaking at 37 °C. The growth of bacteria was assessed by measuring the absorbance at 600 nm every 30 min for 24 h. All experiments were conducted in triplicate. To assess growth kinetics, the standard deviation was calculated for the mean growth kinetic values of each condition. DMFit software was used to calculate the maximum growth rate [21].

### 2.3. Measurement of Biofilm Formation

Two hundred microliters of the *C. jejuni* 11168 cells, which were adjusted to an OD of 0.09 at 600 nm, were carefully transferred to each well of the 96-well plates [22]. The wells contained either CJ, a combination of CJ and MH broth, or BHI broth. Plates were sealed with parafilm and then placed inside the DG250 Microaerophilic Workstation. Following a 72 h incubation period at 37 °C in static conditions, the nonadherent planktonic cells were discarded, and the attached biofilm cells were washed three times using distilled water. Cells were subsequently stained with 125 µL of a 0.1% solution of crystal violet (CV) in water (Thermo Fisher Scientific) for 10 min at room temperature. The excess, unbound CV was then eliminated, and the wells were washed once again with distilled water and air-dried at room temperature for 5 h. CV that bound to the biofilms was solubilized by adding 125 μL of 30% acetic acid (Thermo Fisher Scientific) in water. The microplate was then incubated at room temperature for 10 min. Finally, the results were quantified by measuring the absorbance at a wavelength of 600 nm using the BioTek plate reader. The biofilm experiments were conducted in triplicate. After determining that there are significant differences among group means using ANOVA, Tukey’s HSD test was performed to find out which specific groups differed from each other.

### 2.4. Electron Microscopy

The planktonic cells of *C. jejuni* NCTC 11168 were cultivated overnight in CJ at 37 °C with vigorous shaking in the DG250 Microaerophilic Workstation. Afterward, the cells were centrifuged, washed with PBS, and fixed using 2.5% glutaraldehyde in sodium cacodylate buffer (pH 7.4, MilliporeSigma). To analyze biofilm cells, Thermanox polyester coverslips (Thermo Fisher Scientific) were positioned within a 6-well plate (Corning Inc.). The cell suspensions in CJ were then introduced to each well of the microplates, allowing the biofilm cells to grow for 72 h at 37 °C. Following that, the cells on the coverslips were washed with PBS and fixed with 2.5% glutaraldehyde. The cellular structure of both *C. jejuni* planktonic and biofilm cells was examined using field emission scanning electron microscopy (FESEM) [22]. Cells were washed three times with PBS for 15 min and subsequently dehydrated using a series of ethanol concentrations (15%, 30%, 50%, 70%, 80%, 90%, 95%, and 100%) for 30 min. Once completely dehydrated, the coverslips were placed into an Autosamdri-815 critical point dryer (Tousimis Research Corp., Rockville, MD, USA) and sealed in its chamber while the coverslips were immersed in ethanol. Once sealed in the chamber, the chamber was flushed with liquid CO_2_ and the ethanol was washed out through several washes with CO_2_. Once CO_2_ had replaced the ethanol, the chamber was heated under pressure to take the CO_2_ through its critical point drying and preserve the cells and coverslip without any drying artifacts. Once pressure was slowly decreased to atmosphere, the chamber was opened and the coverslips were mounted onto 10 mm SEM stubs using double-sided conductive carbon tape (Electron Microscopy Sciences, Hatfield, PA, USA). The edges of the glass coverslips were painted with silver paint in order to make a conductive contact between the glass and carbon tape. The stubs were then placed into a Denton Desktop V sputter coater (Denton Vacuum, Moorestown, NJ, USA) where the samples were sputtered for 45 s with a gold–palladium coating to make the samples conductive under the beam. Once the samples were prepared, they were placed into a Zeiss Merlin Gemini2 field emission scanning electron microscope (Carl Zeiss Microscopy, Thornwood, NY, USA). Images were then recorded under different imaging conditions for sample evaluation.

In preparation for transmission electron microscopy (TEM), the planktonic and biofilm cells were rinsed with PBS. *C. jejuni* cells were then fixed with 2.5% glutaraldehyde at 4 °C and subsequently postfixed in 1% osmium tetroxide (Thermo Fisher Scientific) in 0.2 M Sorenson’s phosphate buffer (MilliporeSigma) for 1 h [23]. The cells underwent a dehydration process using a series of ethanol washes, starting with lower concentrations and gradually increasing to 100%. Once fully dehydrated, the cells were then immersed in propylene oxide (MilliporeSigma) to ensure complete dehydration and miscibility with unpolymerized embedding resin. Infiltration with Epon/Araldite resin (Electron Microscopy Sciences) was achieved by gradually introducing a mixture of Epon/Araldite embedding medium and propylene oxide (MilliporeSigma), starting with a 1:2 ratio, followed by a 2:1 ratio, and finally using 100% epoxy embedding medium. The resin was then polymerized by placing the samples in a 60 °C oven overnight. Polymerized cells were sliced into sections using a Leica UC-6 ultramicrotome (Leica Biosystems, Buffalo Grove, IL, USA), and the resulting sections were transferred onto 200 mesh copper grids (Thermo Fisher Scientific). To enhance visibility, the ultrathin sections were stained with uranyl acetate (MilliporeSigma) and lead citrate (MilliporeSigma). The samples were observed using a JEM-2100 200 keV instrument (JEOL USA, Inc., Peabody, MA, USA) operated at an accelerated voltage of 80 keV.

### 2.5. Protein Extraction

The planktonic cells were first calibrated to an OD of 0.09 at 600 nm in 30 mL of CJ. Following calibration, these cells were cultured for 16 h at 37 °C with vigorous shaking. In contrast, the biofilm cells were transferred to a sterile Petri dish (Thermo Fisher Scientific) containing a stainless coupon 304 (Biosurface Technologies, Bozeman, MT, USA). The biofilm cells were then allowed to grow for 72 h at 37 °C within the DG250 Microaerophilic Workstation. Planktonic cells cultured for 16 h were chosen since this period is critical as it allows us to observe the cellular responses and protein expression profiles during a time when the cells are actively growing and metabolizing nutrients. On the other hand, biofilm cells were cultured for 72 h to ensure the formation of a mature biofilm. A 72 h culture period is sufficient to provide a comprehensive view of the proteomic changes associated with biofilm maturation and stability. Collected planktonic and biofilm cells were introduced into Lysing Matrix B tubes (MP Biomedicals, Santa Ana, CA, USA) containing silica beads [24]. Subsequently, 100 µL of BugBuster Plus Lysonase Kit (MilliporeSigma) was carefully added to the tube, and the cells were subjected to disruption using an FP120 reciprocator (MP Biomedicals) at speed 6 for 45 s. Following cell disruption, the mixture was boiled and then vortexed for 5 min and 1 min, respectively. The final protein extract was obtained by centrifuging the disrupted cells at 20,817× *g* for 30 min at 4 °C. The concentration of the extracted protein was determined using a Micro BCA Protein Assay Kit (Thermo Fisher Scientific).

### 2.6. Ultra-High-Performance Liquid Chromatography–Tandem Mass Spectrometry (UHPLC-MS/MS)

Proteomic analysis was conducted by Bioproximity (Bioproximity, LLC, Chantilly, VA, USA). The protein sample was reconstituted using a mixture of 5% SDS, 50 mM Tris-HCl (pH 8.0), 5 mM Tris (2-carboxyethyl) phosphine, and 20 mM 2-chloroacetamide purchased from MilliporeSigma. Afterward, the sample was subjected to digestion using the single-pot solid-phase-enhanced sample preparation method [25]. The digested sample was then analyzed using ultra-high-performance liquid chromatography–tandem mass spectrometry (UHPLC–MS/MS) (Thermo Fisher Scientific) [26]. The LC analysis was performed using an EASY-nLC 1200 system (Thermo Fisher Scientific) coupled to a Q-Exactive HF-X quadrupole-Orbitrap mass spectrometer (Thermo Fisher Scientific). MS settings included a top 12 ion selection and a scan range of 350 to 1400 *m*/*z*. Normalized collision energy was set at 27, automatic gain control was set to 3 × 10^6^, maximum fill MS was set to 45 ms, and maximum fill MS/MS was set to 22 ms. MS1-based isotopic features were detected, and peptide peak areas were calculated using OpenMS (http://dx.doi.org/10.1186/1471-2105-9-163) (accessed on 23 May 2023) [27]. Proteins were considered valid if they had at least one unique peptide identified across all analyzed samples, with E-value scores of 0.0001 or lower. They were classified as differentially expressed if the fold ratio between the control (planktonic cells) and treated groups (biofilm cells) was ≥2.0 (up) or ≤0.5 (down) (|Log_2_ fold ratio| > 1.0). To determine the protein functions, the Cluster of Orthologous Groups (COG) classification system [28] was utilized, while the potential pathways were mapped using the Kyoto Encyclopedia of Genes and Genomes (KEGG) database [29]. Protein interaction network analysis was performed using the Search Tool for the Retrieval of Interacting Genes (STRING) database version 12.0 and Cytoscape version 3.9.1 [30,31].

## 3. Results

### 3.1. Growth Rate and Biofilm Formation in CJ

*C. jejuni* NCTC 11168 cells displayed the high growth rate when cultured in CJ and 1:1 mixtures of CJ with BHI or MH broth (Figure 1). In contrast, the growth of *C. jejuni* cells was slower when cultured in BHI or MH broth alone. The maximum growth rates (h^−1^) were determined as 0.163, 0.075, and 0.069 in CJ and 1:1 mixtures of CJ with BHI and MH broth, respectively. Conversely, the maximum growth rates (h^−1^) in BHI and MH broth alone were calculated as 0.011 and 0.005. The extent of biofilm formation exhibited a similar pattern to the growth rate of planktonic cells (Figure 2). Notably, significantly more biofilms were produced in CJ.

### 3.2. Microscopic Image Analysis by FESEM and TEM

Through FESEM analysis, an abundance of curved, rod-shaped cells was observed in the planktonic cells grown in CJ (Figure 3A). However, as depicted in Figure 3B, we also detected elongated cells, exceeding 13 µm in length. FESEM analysis clearly demonstrated the remarkable formation of biofilms by *C. jejuni* after 72 h of cultivation (Figure 4A–C). The morphological characteristics of *C. jejuni* biofilms exhibited significant disparities when compared to their planktonic counterparts. FESEM images showed the presence of dense, multilayered bacterial cells adhering to one another, forming thick and fully established three-dimensional biofilms. While coccoid-shaped cells were abundant in the biofilms, helical-shaped cells were also observed (Figure 4). While some biofilms were characterized by thick, compact structures (Figure 4A,B,E,H), others exhibited more open and porous matrices with less dense networks (Figure 4C,D,F). A conditioning layer, which provides an environment suited for biofilm formation, was also observed (Figure 4C). Remarkably, *C. jejuni* biofilm cells formed numerous extended thread-like structures, reaching lengths exceeding 40 μm (Figure 4A,D, yellow arrows). These structures were notably thicker than flagella.

Unusually elongated spiral-shaped cells, similar to planktonic cells, were detected within the biofilms (Figure 4E, red arrow). Additionally, we occasionally observed thinner stretched structures (Figure 4F–H, blue arrows). Figure 4G,H exhibited biofilms with numerous small, rounded coccoid cells, forming a compact network. To our knowledge, the presence of these unique structures in *C. jejuni* biofilms has not been previously reported. TEM images further confirmed the predominance of coccoid-shaped cells in the biofilms, with limited presence of vegetative rod-shaped cells (Figure 5).

### 3.3. Proteome Profile of C. jejuni Biofilms in Chicken Juice after 72 h of Incubation

Upon examining the global proteome data of *C. jejuni* NCTC 11168 biofilms formed in CJ after 72 h of incubation, a total of 1441 proteins were identified. Among them, 271 exhibited upregulation, while 663 showed downregulation (Figure 6, Appendix A). In each COG category, more proteins were downregulated than were upregulated (Figure 7). “Metabolism” was the most prevalent category among both upregulated (80) and downregulated (294) proteins, with “Cellular processes and signaling” ranking second (74 up, 168 down), followed by “Information storage and processing” (52 up, 104 down). Except for the category “Function unknown”, the primary COGs associated with upregulated proteins were “Cell wall/membrane/envelope biogenesis” (M, 29), “Translation, ribosomal structure and biogenesis” (J, 26), and “Amino acid transport and metabolism” (E, 18) (Figure 8). Conversely, underexpressed proteins were predominantly associated with “Energy production and conversion” (C, 71), “Amino acid transport and metabolism” (E, 64), and “Translation, ribosomal structure and biogenesis” (J, 58).

Like COGs, within each KEGG category, more proteins were downregulated than were upregulated (Figure 9, Appendix A). Among the five primary KEGG categories (Metabolism, Cellular processes, Environmental information processing, Genetic information processing, and Human diseases), “Metabolism” had the highest number of both upregulated (189) and downregulated (839) proteins, followed by “Genetic Information Processing” (27 up, 75 down). The smallest categories were “Environmental Information Processing” (15 up, 56 down) and “Human Diseases” (0 up, 8 down). Specifically, within the “Metabolism” group, excluding the “Global and overview map” category, the subcategories of “Metabolism of cofactors and vitamins”, “Amino acid metabolism”, and “Carbohydrate metabolism” exhibited the highest numbers of proteins in differentially expressed proteins (Appendix A). Notably, “Metabolic pathways” was the most affected category, with both the most upregulated (53) and downregulated (210) proteins (Figure 10A,B). Several other pathways related to basic cellular processes were also significantly affected, including “Biosynthesis of secondary metabolites” (22 up, 98 down), “Biosynthesis of cofactors” (18 up, 48 down), “Biosynthesis of amino acids” (10 up, 42 down), and “Microbial metabolism in diverse environments” (8 up, 49 down).

Our proteomic analysis of *C. jejuni* strain 11168 biofilm in CJ identified 48 (82.8%) of the 58 known biofilm-associated proteins. Interestingly, 30 (62.5%) of these proteins exhibited differential expression, with three specifically upregulated and the remainder downregulated. These differentially expressed proteins were associated with such diverse functions as quorum sensing, motility, stress response, chemotaxis, and surface structures (Table 1). The ABC transporter ATP-binding protein (Cj0732) showed the most significant upregulation with a Log_2_ fold ratio of 24.90, followed by alkaline phosphatase (PhoX) with a Log_2_ fold ratio of 3.14, and ABC transporter substrate-binding protein (LivJ) with a Log_2_ fold ratio of 1.80. Among downregulated proteins, oligosaccharyltransferase (PglB) showed the greatest decrease (Log_2_ fold change: −31.08), followed by ABC transporter substrate-binding protein (LivK, −28.81) and signal recognition particle protein (FtsY, −28.20). Analysis of differentially expressed proteins using Cytoscape unveiled distinct clusters in the resulting protein–protein network (Figure 11A). A prominent cluster of encompassed proteins was implicated in flagellar assembly and signal transduction, with CheA (histidine kinase sensor) emerging as a central hub interacting with diverse partners, including AhpC, CheW, Cj1110c, Cj1564, Ffh, FlaG, FlhA, FliA, FliS, PflA, and SpoT.

Our proteome analysis revealed a total of 106 virulence proteins with differential expression, with 31 upregulated and the remaining 75 downregulated (Table 2). Major virulence groups included proteins associated with flagellar assembly (FlgB-D, FlgG-J, FlhA, FliA, FliD-E, FliG, FliI, FliL-N, FliS, MotA-B, RpoN), bacterial chemotaxis (CheA, CheW, Cj1564, FliG, FliM-N, MotA-B), lipopolysaccharide (LPS) biosynthesis (GmhB, HddA, HddC, HldD, HtrB, LpxB, KdsA-B, KpsF, WaaC, WaaF), biosynthesis of nucleotide sugar (Glf, GmhB, Gne, HddA, HddC, HldD, KdsA-B, KfiD, KpsF, NeuB1-C1, PglE, PseB-C, PseF-H, RfbC), O-antigen nucleotide sugar biosynthesis (Glf, Gne, KfiD, NeuB1-C1, PseB-C, PseF-H, RfbC), amino sugar and nucleotide sugar metabolism (Glf, Gne, KfiD, NeuB1-C1, PglE, PseB-C, PseF-H), and two-component system (CheA, CheW, Cj1564, FliA, HtrA, MotA, RpoN). Following Cytoscape analysis, it was evident that one significant cluster comprised proteins involved in flagellar assembly and bacterial chemotaxis (Figure 11B). Within this cluster, CheA (histidine kinase sensor), FliM (flagellar motor switch protein), and FlhA (flagellar biosynthesis protein) acted as central hubs, interacting with over 30 proteins. Additionally, another major cluster comprised proteins involved in LPS and nucleotide sugar biosynthesis. Notably, RfbC (dTDP-4-dehydrorhamnose 3,5-epimerase), Cj1421c (sugar transferase), and KpsF (D-arabinose 5-phosphate isomerase) emerged as central hubs, interacting with more than 26 proteins.

*C. jejuni* has demonstrated that iron metabolism and acquisition play a role in the biofilm formation process [32]. Among the proteins analyzed, five were upregulated, while thirteen were downregulated (Table 3). Notably, multiple iron acquisition and transport systems were differentially expressed, including energy-transduction systems (ExbD1, TonB1-B2), ferric enterochelin uptake (CeuC-E, CfrA), haemin uptake (ChuA-C), and siderophore uptake (Cj1658, Cj1661, Cj1663). Cytoscape analysis identified multiple iron acquisition and transport systems clustered together, with proteins like Cj1658 (iron permease) and ChuA (haemin uptake system) acting as central hubs (Figure 11C).

## 4. Discussion

Poultry meat serves as the main source of *C. jejuni* infection, playing a crucial role in its transmission and contributing significantly to the prevalence of gastrointestinal infections caused by this pathogen [7]. Most studies investigating *C. jejuni* biofilms were carried out in laboratory settings that failed to accurately mimic the actual conditions found in food processing environments. Our study aimed to facilitate the accurate extrapolation of laboratory findings to the food industry. CJ closely resembles the natural environment of *C. jejuni* and provides a suitable laboratory model for studying biofilm formation in the food chain [18,20,33]. Its use as a growth medium provides a more accurate simulation of the conditions that lead to the dissemination and cross-contamination of *C. jejuni* in chicken carcasses.

Previous reports demonstrated that *C. jejuni* exhibits longer viability in CJ than in BHI broth at both refrigerated and freezing temperatures (5 °C, 10 °C, −18 °C) [20,33]. In addition, agar plates supplemented with CJ supported the growth of multiple clinical isolates and a chicken-derived *C. jejuni* strain at both 37 °C and 42 °C. We observed a high growth rate of *C. jejuni* when cultivated in CJ and BHI or MH broth supplemented with CJ. This represents the first detailed examination of *C. jejuni*’s growth dynamics in CJ at optimal conditions. Taken together, these findings strongly suggest that CJ significantly enhances *C. jejuni* growth, potentially impacting its persistence and survival in relevant environments.

Brown et al. investigated the impact of CJ on the formation of *C. jejuni* biofilms [18]. Their findings revealed that the presence of CJ resulted in an increase in biofilm formation, which was attributed to enhanced attachment to abiotic surfaces rather than simply to an increase in cell numbers. Our study aligns with these previous findings, as we also observed significantly higher levels of biofilm formation in CJ and laboratory media supplemented with 50% CJ. In Brown et al., *C. jejuni* biofilm cells were grown in CJ at 37 °C for 48 h [18]. They observed that *C. jejuni* biofilm cells showed a preference for binding to CJ particulates rather than directly to Thermanox coverslips. Moreover, they observed that CJ created a conditioning layer on abiotic surfaces, facilitating biofilm formation by *C. jejuni*. Similarly, in our own investigations, we also observed a tendency for *C. jejuni* biofilm cells to attach to CJ particulates and the conditioning layer.

Previous studies have shown that under various stress conditions, including suboptimal temperature, oxidative stress, nutrient limitation, and alterations in pH and osmotic pressure, *C. jejuni* undergoes morphological changes, transitioning from its characteristic helical form to a coccal shape [34,35]. These coccoid cells were reported to be metabolically inactive but to remain viable. We found that *C. jejuni* biofilm formed after 72 h at 37 °C was mostly coccoid-shaped, although some regions harbored spiral cells. This observation aligns with findings showing that most cells in the biofilm of *Helicobacter pylori* are cocci forms in vitro [36]. Furthermore, biopsy samples obtained from patients revealed that coccoid cells of *H. pylori* formed biofilms attached to the gastric mucosa, protected by the matrix [37]. Nutrient depletion due to prolonged culture period is a potential explanation for this morphological shift. *C. jejuni* biofilm cells exhibited remarkable morphological features, including extended thread-like structures and thinner stretched structures. These previously undescribed structures are presumed to contribute significantly to the maintenance of *C. jejuni* biofilms.

While previous efforts have focused on identifying genes associated with *C. jejuni* biofilm formation in laboratory media [38,39], our study leverages global proteomics to pioneer the identification of protein players driving *C. jejuni* biofilm formation in CJ. Both COG and KEGG pathway analyses revealed a higher number of downregulated proteins compared to upregulated ones in our *C. jejuni* biofilms cultured in CJ. This aligns with the predominance of coccoid cells observed in FESEM images, as these forms are known to have reduced metabolic activity compared to their actively growing and spiral counterparts [34,35], This suggests a potential decrease in overall protein synthesis activity within the coccoid-dominated biofilm.

This proteome study uncovers a prominent role for “Cell wall/membrane/envelope biogenesis” proteins in mature biofilm formed in CJ. Notably, upregulated proteins within this COG functional group outnumbered any other, highlighting their crucial contribution to maintaining biofilm structure and stability. Interestingly, Cj0093 (putative periplasmic protein), RlpA (endolytic peptidoglycan transglycosylase), Cj1137c (glycosyltransferase), and PldA (phosphatidylcholine 1-acylhydrolase) emerged as the most significantly upregulated members. This finding mirrors similar observations by Lopez-Fernandez et al. in *Thiobacillus* biofilms, where “Cell wall/membrane/envelope biogenesis” reigned supreme among overexpressed COG categories [40]. Additionally, Benamara et al. demonstrated significant changes in membrane composition in *P. aeruginosa* biofilms, further accentuating the importance of this cellular component [41]. These observations, collectively, paint a compelling picture. Proteins associated with the outer membrane, the periplasm, and the inner membrane appear to be critical for maintaining biofilm integrity, aligning with previous reports on their roles in Gram-positive and -negative bacteria [42,43].

Distinct proteomic alterations were noted in ribosomal proteins within *C. jejuni* biofilm cells in CJ. Specifically, 15 proteins showed downregulation while 10 exhibited upregulation, with 8 and 17 proteins belonging to the 30S and 50S ribosomal proteins, respectively. Compared to planktonic cells, biofilm cells displayed reduced protein synthesis activity, suggesting a potentially modulated metabolic state with diminished, yet not entirely inactive, protein production. Intriguingly, Khan et al. reported significant overexpression of ribosomal proteins in *Burkholderia thailandensis* biofilms [44]. This discrepancy potentially stems from differing biofilm formation methods. While our study employed static incubation for 72 h, Khan et al. utilized a continuous nutrient supply through a flow reactor system. Under our static conditions, the predominance of coccoid cells with reduced metabolic activity may be a result of limited upregulation in ribosomal protein synthesis, unlike the actively growing cells in Khan et al.’s study, highlighting the influence of environmental factors on biofilm protein expression.

While quorum sensing is known to be crucial for biofilm formation [45], its dynamics often remain complex and context-dependent. In this investigation, we explored changes in *C. jejuni* biofilms grown to late stage in CJ, and identified significant alterations in 11 proteins associated with quorum sensing compared to planktonic growth. Among these, RibA, YajC, FtsY, and LivK exhibited the most substantial downregulation with Log_2_ fold ratios ranging from −23.69 to −28.81. Interestingly, LuxS, despite its well-known role in regulating biofilm formation, showed no significant alteration, with a Log_2_ fold ratio of −0.31 [46]. These findings align with the established role of quorum sensing in the early stages of biofilm development, as shown by Rampadarath et al. [47]. However, it is worth noting that some studies have shown differential regulation of quorum-sensing proteins in *S. aureus* and *B. longum* biofilms grown for 72 and 86 h, respectively, with some being upregulated while others are downregulated [48], highlighting the complexity of this regulatory system.

Flagella, the whip-like appendages powering bacterial motility, are known to facilitate bacterial adherence and movement towards nutrients [49]. They are particularly essential in the early stages of biofilm formation, aiding bacteria in locating and interacting with suitable surfaces [50]. For instance, mutations in key flagellar components, like regulators (*flhD*) and structural proteins (*fliC*), significantly hinder biofilm formation in some bacteria [51]. Similarly, disruptions in motility and flagellar assembly proteins (*motA*, *motB*, *flgF*, *flhB*, *fleS*) also had detrimental effects [51,52]. In our investigation, we observed downregulation of 15 flagella assembly proteins, with particularly drastic suppression of FlgH, MotA, and FliE (Log_2_ fold ratios ranging from −25.29 to −29.45) in mature *C. jejuni* biofilms. These findings suggest that while flagella are critical in the early stages of biofilm formation, their significance may diminish in later stages.

Iron, a vital micronutrient, is known to influence biofilm formation, virulence, and even antimicrobial resistance in bacteria [53]. Interestingly, our investigation into late-stage biofilms identified downregulation of 13 proteins associated with iron uptake. Notably, seven of these proteins exhibited pronounced suppression, with Log_2_ fold ratios ranging from −23.75 to −28.21. These findings align with observations in other bacterial species like *A. baumannii* and *Pseudomonas aeruginosa*, in which mature biofilms (both four days old) displayed repression of numerous iron uptake proteins [53,54]. These findings collectively suggest a diminished iron uptake potential in late-stage biofilms across diverse bacterial species.

We identified a significant number of differentially expressed proteins associated with the stress response. Among these, the ATP-dependent Clp protease proteolytic subunit protein (ClpP) is known to play a crucial role in bacterial growth and serves as a key component in cellular protein quality control systems, involved in refolding or degrading damaged proteins [55]. Interestingly, various studies have implicated ClpP in biofilm formation across various Gram-positive and -negative pathogens, including *Actinobacillus pleuropneumoniae*, *Haemophilus parasuis*, *P. aeruginosa*, *Porphyromonas gingivalis*, *S. aureus*, *S. epidermidis*, and *Streptococcus mutans* [56,57,58,59,60,61,62]. Notably, our proteome analysis revealed a significant downregulation of ClpP with a Log_2_ fold change of −27.93. Additionally, Gundogdu et al. demonstrated that the MarR-type transcriptional regulator RrpA regulates *C. jejuni*’s oxidative and aerobic stress responses, enhancing survival [63]. Mutations in this gene altered biofilm formation, and our analysis revealed a significant upregulation of RrpA (Log_2_ 27.18-fold increase). These combined findings suggest potential roles for both ClpP and RrpA in regulating *C. jejuni* biofilm formation. Beyond ClpP and RrpA, our analysis identified other significantly altered stress-responsive proteins, including a membrane protein (Cj1623, 23.97-fold increase) and a DnaJ-class molecular chaperone protein (CbpA, −29.86 fold decrease). Given the lack of existing research on these proteins in the context of *C. jejuni* biofilm formation, further investigation to elucidate their potential roles is warranted.

While this study sheds light on protein expression in late-stage biofilms in CJ, acknowledging its limitations is crucial to gaining a more complete picture. Analyzing only mature stages restricts our grasp of the dynamic protein changes throughout biofilm development. Additionally, studying biofilms solely in CJ limits the generalizability of our findings. Comparing protein profiles from diverse media, such as MH and BHI broths, could reveal how bacteria adapt to different nutrient environments and potentially unveil media-specific biofilm characteristics. 

## 5. Conclusions

*C. jejuni*, a notorious foodborne pathogen, flourishes in CJ, forming robust biofilms with intriguing morphologies. Global proteomic analysis revealed a predominance of downregulated proteins, hinting at reduced metabolic activity in mature biofilms. Specifically, proteins associated with ribosomal function, amino acid, purine, and pyrimidine metabolism, quorum sensing, iron uptake, and flagellar assembly exhibited diminished expression, aligning with the observed shift towards a coccoid morphology. This study lays the groundwork for further exploration of *C. jejuni* biofilm formation, ultimately leading to the development of more effective control strategies for combatting this pathogen.

## Figures and Tables

**Figure 1 foods-13-01828-f001:**
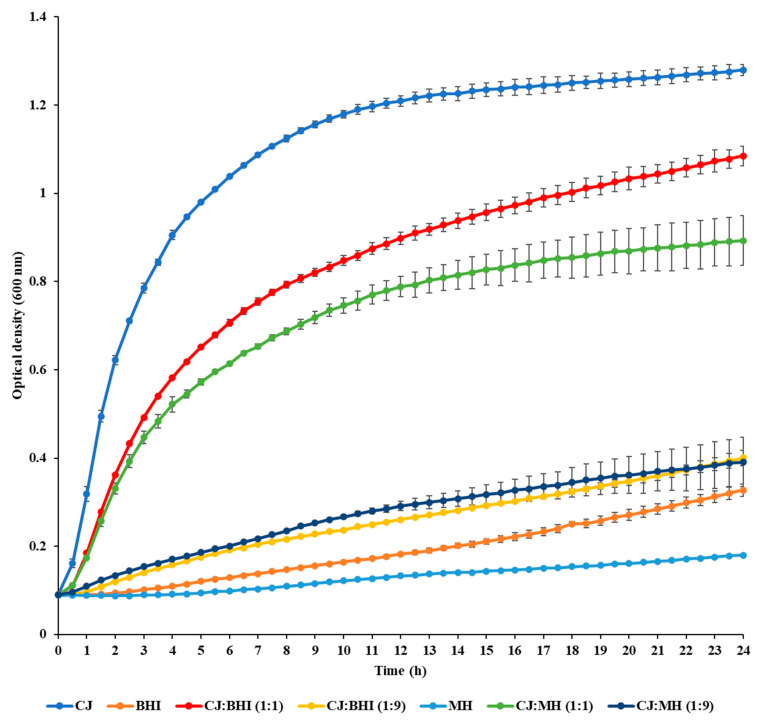
Growth curve of *C. jejuni* strain 11168 in different growth media for 24 h at 37 °C. **CJ**: chicken juice, **BHI**: brain heart infusion, **MH**: Mueller Hinton, **CJ:BHI (1:1)**: 1:1 mixture of CJ and BHI, **CJ:BHI (1:9)**: 1:9 mixture of CJ and BHI, **CJ:MH (1:1)**: 1:1 mixture of CJ and MH, **CJ:MH (1:9)**: 1:9 mixture of CJ and MH. Experiments were performed in triplicate to calculate the mean.

**Figure 2 foods-13-01828-f002:**
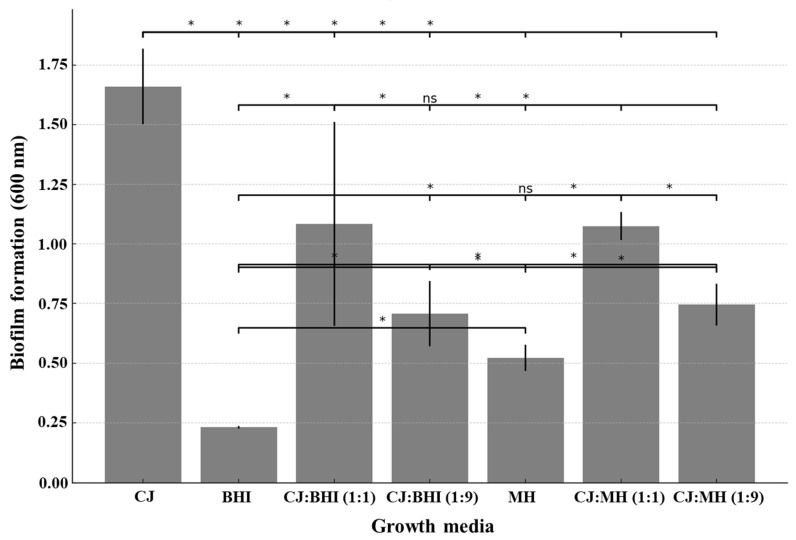
Biofilm formation of *C. jejuni* strain 11168 in different growth media for 72 h at 37 °C. **CJ**: chicken juice, **BHI**: brain heart infusion, **MH**: Mueller Hinton, **CJ:BHI (1:1)**: 1:1 mixture of CJ and BHI, **CJ:BHI (1:9)**: 1:9 mixture of CJ and BHI, **CJ:MH (1:1)**: 1:1 mixture of CJ and MH, **CJ:MH (1:9)**: 1:9 mixture of CJ and MH. Experiments were performed in triplicate to calculate the mean and standard deviation. Statistical significance between groups is indicated with asterisks (*) for significant differences and “ns” for not significant.

**Figure 3 foods-13-01828-f003:**
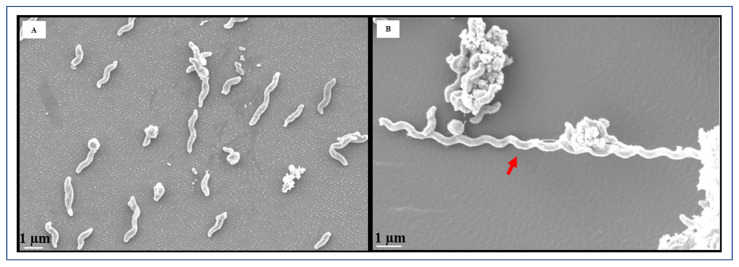
Field emission scanning electron microscopy (FESEM) images of *C. jejuni* strain 11168 planktonic cells grown in chicken juice overnight at 37 °C. (**A**) Typical planktonic cells; (**B**) Planktonic cells showing elongated cells. The scale bar in all the images corresponds to 1 µm. Red arrow indicates elongated cells.

**Figure 4 foods-13-01828-f004:**
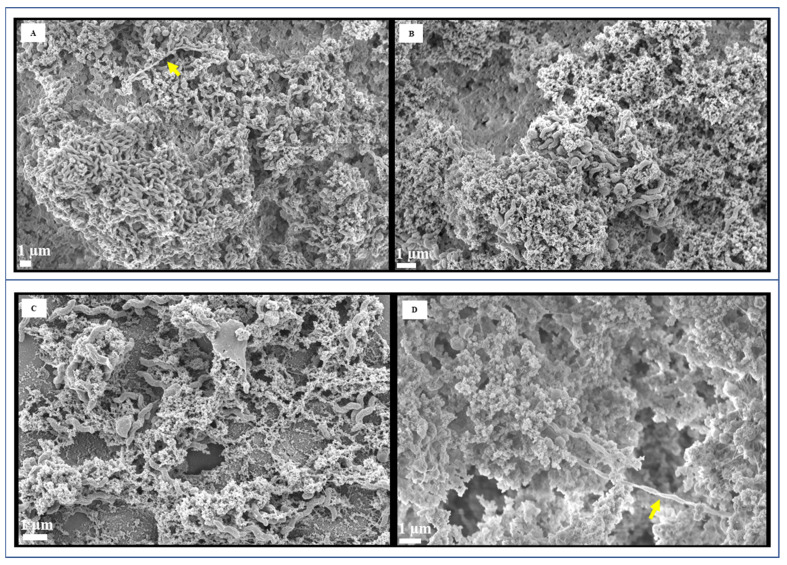
Field emission scanning electron microscopy (FESEM) images of *C. jejuni* strain 11168 biofilms grown in chicken juice for 72 h at 37 °C. (**A,B**) A dense network of biofilm with rough surface morphology; (**C**) A more open and porous biofilm structure; (**D**) An open biofilm structure, but with noticeable filamentous structures; (**E**) A dense biofilm with some distinguishable filamentous components; (**F**) A more open structure with numerous filamentous connections; (**G**) A biofilm with many small, rounded cellular components; (**H**) A dense biofilm with both filamentous and rounded components. The scale bar in all the images corresponds to 1 µm. Yellow arrows: extended thread-like structures; red arrow: elongated spiral-shaped; blue arrows: thinner stretched structures.

**Figure 5 foods-13-01828-f005:**
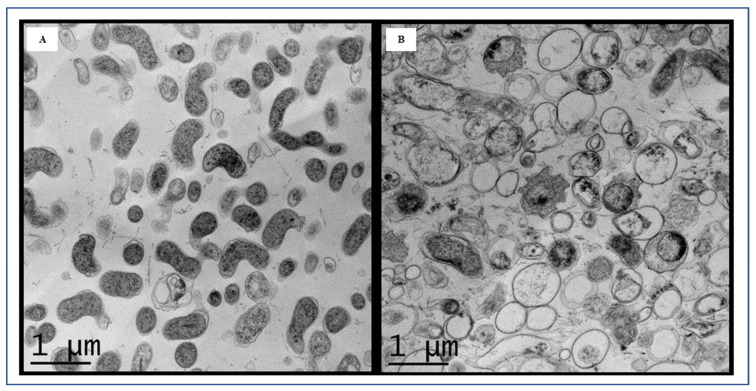
Transmission electron microscopy (TEM) images of *C. jejuni* strain 11168 planktonic (**A**) and biofilm (**B**) cells grown in chicken juice. The scale bar in all the images corresponds to 1 µm.

**Figure 6 foods-13-01828-f006:**
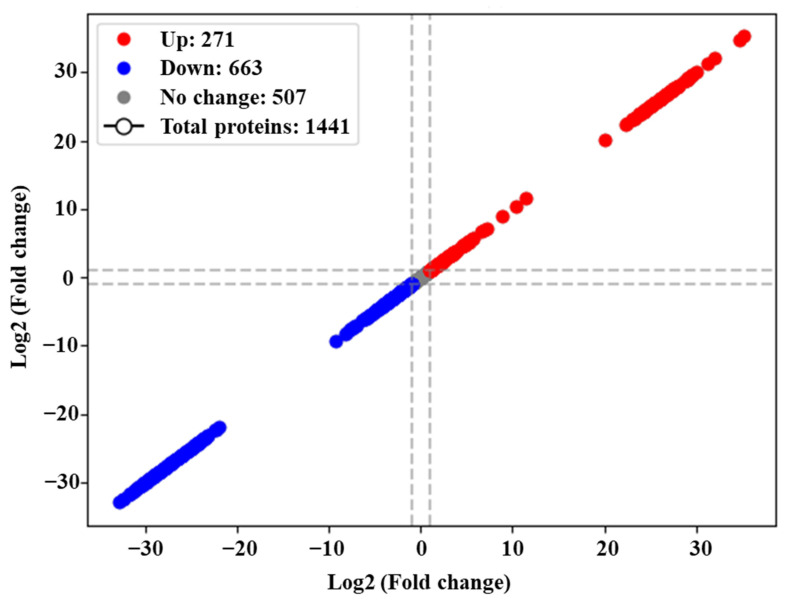
Distribution of the total identified proteins of *C. jejuni* strain 11168 biofilms in chicken juice. The *X*- and *Y*-axes show Log_2_ fold change groups.

**Figure 7 foods-13-01828-f007:**
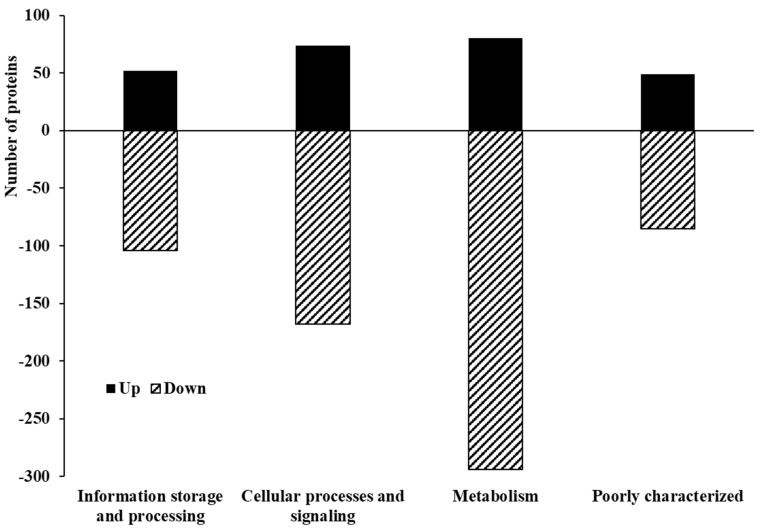
Cluster of Orthologous Groups (COG) functional classification of four major categories in differentially expressed proteins identified from *C. jejuni* strain 11168 biofilms in chicken juice.

**Figure 8 foods-13-01828-f008:**
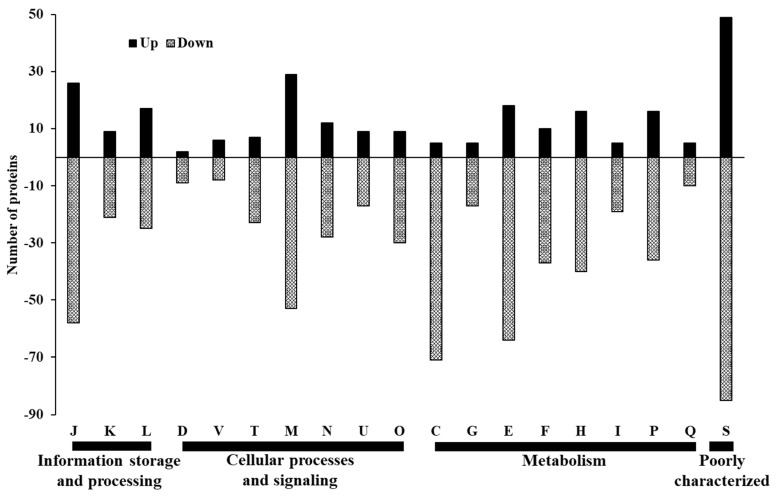
Detailed Cluster of Orthologous Groups (COG) functional classification in differentially expressed proteins identified from *C. jejuni* strain 11168 biofilms in chicken juice. COG functional categories: **J**, translation, ribosomal structure, and biogenesis; **A**, RNA processing and modification; **K**, transcription; **L**, replication, recombination, and repair; **B**, chromatin structure and dynamics; **D**, cell cycle control, cell division, chromosome partitioning; **Y**, nuclear structure; **V**, defense mechanisms; **T**, signal transduction mechanisms; **M**, cell wall/membrane/envelope biogenesis; **N**, cell motility; **Z**, cytoskeleton; **W**, extracellular structures; **U**, intracellular trafficking, secretion, and vesicular transport; **O**, posttranslational modification, protein turnover, chaperones; **C**, energy production and conversion; **G**, carbohydrate transport and metabolism; **E**, amino acid transport and metabolism; **F**, nucleotide transport and metabolism; **H**, coenzyme transport and metabolism; **I**, lipid transport and metabolism; **P**, inorganic ion transport and metabolism; **Q**, secondary metabolite biosynthesis, transport, and catabolism; **R**, general function prediction only; **S**, function unknown.

**Figure 9 foods-13-01828-f009:**
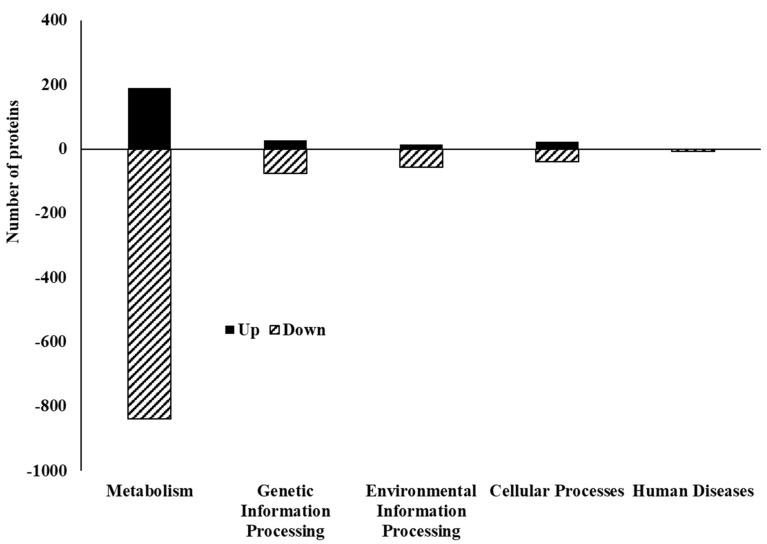
Kyoto Encyclopedia of Genes and Genomes (KEGG) pathways of five major groups in differentially expressed proteins identified from *C. jejuni* strain 11168 biofilms in chicken juice.

**Figure 10 foods-13-01828-f010:**
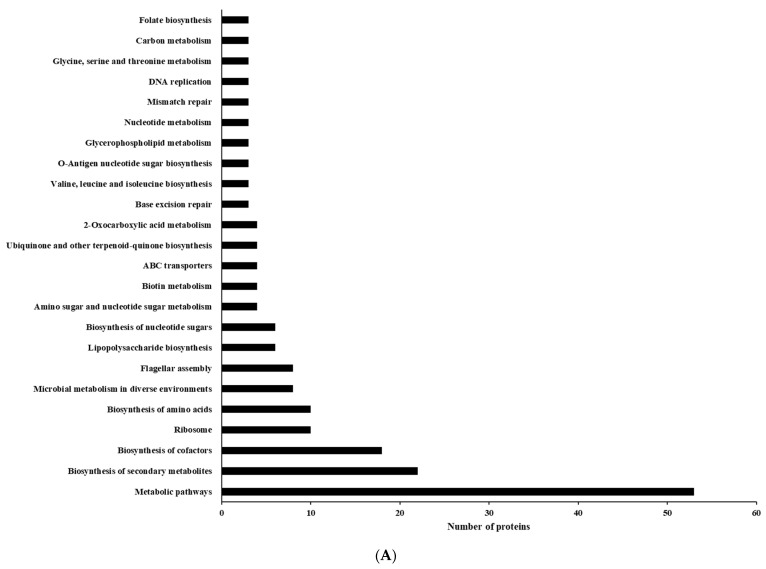
Top 24 Kyoto Encyclopedia of Genes and Genomes (KEGG) pathways in differentially expressed proteins identified from *C. jejuni* strain 11168 biofilms in chicken juice. (**A**) Upregulated proteins; (**B**) Downregulated proteins.

**Figure 11 foods-13-01828-f011:**
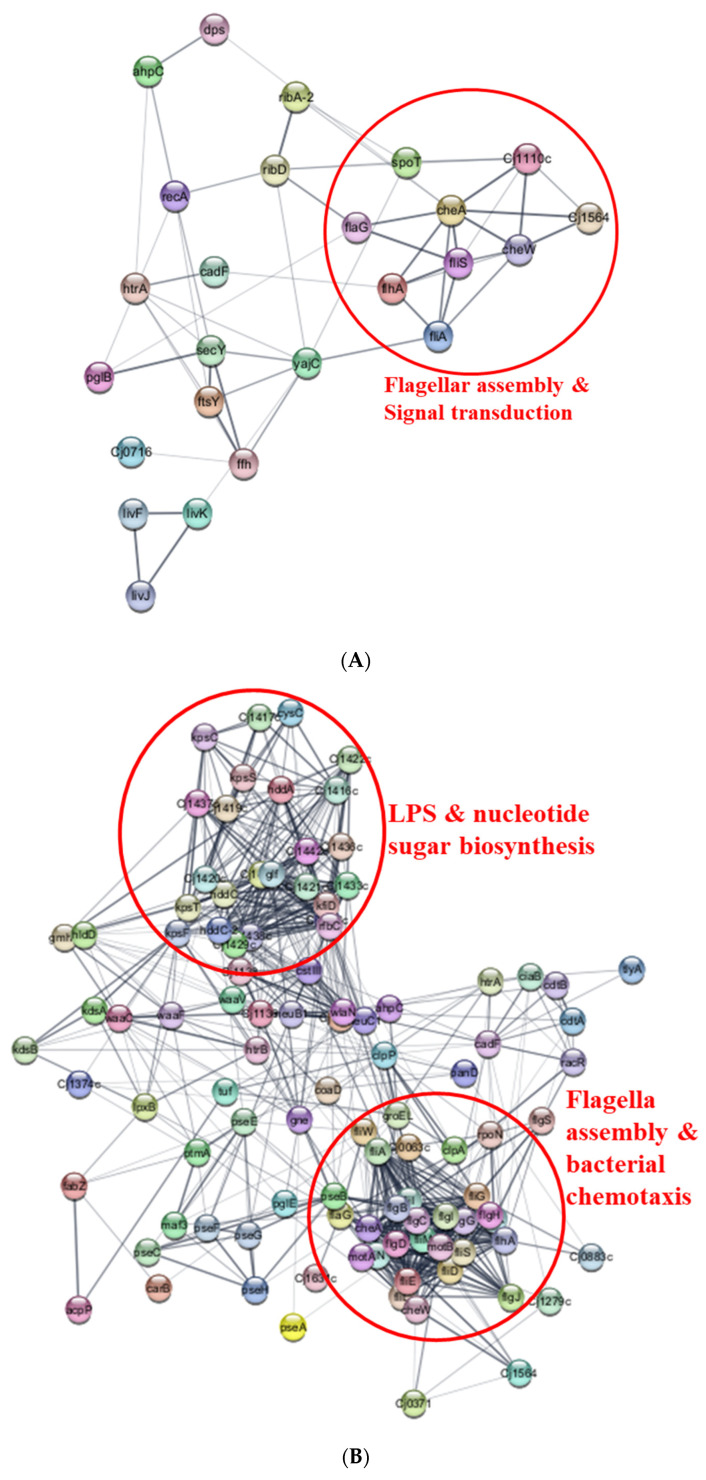
Protein–protein interaction network analysis of differentially expressed proteins. (**A**) Biofilms; (**B**) Virulence; (**C**) Iron metabolism and acquisition.

**Table 1 foods-13-01828-t001:** Differentially expressed proteins associated with biofilm identified from *C. jejuni* strain 11168 biofilms in chicken juice. COG: Cluster of Orthologous Groups. **E**, amino acid transport and metabolism; **F**, nucleotide transport and metabolism; **H**, coenzyme transport and metabolism; **K**, transcription; **L**, replication, recombination, and repair; **M**, cell wall/membrane/envelope biogenesis; **N**, cell motility; **O**, posttranslational modification, protein turnover, chaperones; **P**, inorganic ion transport and metabolism; **S**, function unknown; **T**, signal transduction mechanisms; **U**, intracellular trafficking, secretion, and vesicular transport.

Locus Tag	Genes	Description	COG	Log_2_ (Fold Ratio)
Cj0061c	*fliA*	Sigma factor 28	K	−2.85
Cj0145	*phoX*	Alkaline phosphatase	S	3.14
Cj0283c	*cheW*	Phosphotransferase	NT	−1.01
Cj0284c	*cheA*	Histidine kinase sensor	T	−1.40
Cj0334	*ahpC*	Alkyl hydroperoxide reductase	O	−1.18
Cj0547	*flaG*	Flagellar filament length control	N	−2.04
Cj0549	*fliS*	Flagellar secretion chaperon	N	−2.94
Cj0604	*ppk-2*	Polyphosphate kinase	S	−1.28
Cj0709	*ffh*	Signal recognition particle protein	U	−2.15
Cj0716		Phospho-2-dehydro-3-deoxyheptonate aldolase	E	−1.11
Cj0732		ABC transporter ATP-binding protein	P	24.90
Cj0882c	*flhA*	Flagellar biosynthesis protein	N	−3.48
Cj0996	*ribA*	GTP cyclohydrolase II	F	−23.69
Cj1014c	*livF*	ABC transporter ATP-binding protein	E	−2.88
Cj1018c	*livK*	ABC transporter substrate-binding protein	E	−28.81
Cj1019c	*livJ*	ABC transporter substrate-binding protein	E	1.80
Cj1094c	*yajC*	Protein translocase subunit	U	−27.29
Cj1110c	*tlp8*	Transducer-like protein-8	NT	−1.26
Cj1126c	*pglB*	Oligosaccharyltransferase	S	−31.08
Cj1148	*waaF*	Heptosyltransferase II	M	−27.57
Cj1206c	*ftsY*	Signal recognition particle protein	U	−28.20
Cj1228c	*htrA*	Serine protease	M	−1.12
Cj1272c	*spoT*	Guanosine-3′,5′-bis(Diphosphate) 3′-pyrophosphohydrolase	KT	−2.29
Cj1478c	*cadF*	Outer membrane fibronectin-binding protein	M	−1.62
Cj1534c	*dps*	DNA protection during starvation protein	P	−1.97
Cj1564	*tlp3*	Transducer-like protein-3	NT	−1.38
Cj1565c	*pflA*	Paralyzed flagellum protein	N	−2.01
Cj1622	*ribD*	Riboflavin-specific deaminase/reductase	H	−3.22
Cj1673c	*recA*	Recombinase A	L	−1.46
Cj1688c	*secY*	Protein translocase subunit SecY	U	−26.61

**Table 2 foods-13-01828-t002:** Differentially expressed proteins associated with virulence identified from *C. jejuni* strain 11168 biofilms in chicken juice. COG: Cluster of Orthologous Groups. **C**, energy production and conversion; **D**, cell cycle control, cell division, chromosome partitioning; **E**, amino acid transport and metabolism; **F**, nucleotide transport and metabolism; **G**, carbohydrate transport and metabolism; **H**, coenzyme transport and metabolism; **I**, lipid transport and metabolism; **J**, translation, ribosomal structure, and biogenesis; **K**, transcription; **L**, replication, recombination, and repair; **M**, cell wall/membrane/envelope biogenesis; **N**, cell motility; **O**, posttranslational modification, protein turnover, chaperones; **P**, inorganic ion transport and metabolism; **Q**, secondary metabolite biosynthesis, transport, and catabolism; **S**, function unknown; **T**, signal transduction mechanisms; **U**, intracellular trafficking, secretion, and vesicular transport.

Locus Tag	Genes	Description	COG	Log_2_ (Fold Ratio)
Cj0042	*flgD*	Flagellar basal-body rod modification protein	N	2.24
Cj0060c	*fliM*	Flagellar motor switch protein	N	−1.85
Cj0061c	*fliA*	RNA polymerase sigma factor for flagellar operon	K	−2.85
Cj0063c		Flagellar synthesis regulator	D	−3.43
Cj0078c	*cdtB*	Cytolethal distending toxin B	S	−28.24
Cj0079c	*cdtA*	Cytolethal distending toxin subunit A	M	−26.86
Cj0192c	*clpP*	ATP-dependent Clp protease proteolytic subunit	O	−27.93
Cj0195	*fliI*	Flagellum-specific ATP synthase	NU	1.83
Cj0273	*fabZ*	3-hydroxyacyl-[acyl-carrier-protein] dehydratase	I	25.14
Cj0279	*carB*	Carbamoyl-phosphate synthase large chain	F	−1.10
Cj0283c	*cheW*	Positive regulator of CheA protein activity	NT	−1.01
Cj0284c	*cheA*	Signal transduction histidine kinase	T	−1.40
Cj0288c	*lpxB*	Lipid-A-disaccharide synthase	M	1.07
Cj0296c	*panD*	Aspartate 1-decarboxylase	H	−28.06
Cj0319	*fliG*	Flagellar motor switch protein	N	−7.39
Cj0334	*ahpC*	Alkyl hydroperoxide reductase subunit C-like protein	O	−1.18
Cj0336c	*motB*	Flagellar motor rotation protein MotB	N	10.41
Cj0337c	*motA*	Flagellar motor rotation protein	N	−29.44
Cj0351	*fliN*	Flagellar motor switch protein	N	23.96
Cj0371		Putative flagellar motility protein	-	−1.70
Cj0384c	*kdsA*	2-dehydro-3-deoxyphosphooctonate aldolase	M	−1.02
Cj0441	*acpP*	Acyl carrier protein	IQ	1.53
Cj0470	*tuf*	Translation elongation factor Tu	J	1.16
Cj0526c	*fliE*	Flagellar hook-basal body complex protein	N	−29.45
Cj0527c	*flgC*	Flagellar basal-body rod protein	N	2.07
Cj0528c	*flgB*	Flagellar basal-body rod protein	N	−3.39
Cj0547	*flaG*	Flagellar protein	N	−2.04
Cj0548	*fliD*	Flagellar cap protein	N	−1.74
Cj0549	*fliS*	Flagellar biosynthesis protein	N	−2.94
Cj0588	*tlyA*	RNA binding methyltransferase FtsJ like	J	−24.14
Cj0670	*rpoN*	RNA polymerase sigma-54 factor	K	−1.62
Cj0687c	*flgH*	Flagellar L-ring protein	N	−25.29
Cj0698	*flgG*	Flagellar basal-body rod protein	N	−1.81
Cj0767c	*coaD*	Phosphopantetheine adenylyltransferase	H	−3.83
Cj0788	*ciaD*	2-oxoglutarate:acceptor oxidoreductase	-	−3.63
Cj0793	*flgS*	Flagellar sensory histidine kinase	T	1.07
Cj0813	*kdsB*	3-deoxy-manno-octulosonate cytidylyltransferase	M	−1.34
Cj0882c	*flhA*	Flagellar biosynthesis protein	N	−3.48
Cj0883c		Rrf2 family transcriptional regulator	K	−1.03
Cj0914c	*ciaB*	Campylobacter invasion antigen B	G	−1.51
Cj1024c	*flgR*	Signal-transduction regulatory protein	T	−2.38
Cj1075	*fliW*	Flagellar assembly factor	N	−1.26
Cj1108	*clpA*	ATP-dependent Clp protease ATP-binding subunit	O	−2.27
Cj1121c	*pglE*	UDP-N-acetylbacillosamine transaminase	E	−2.45
Cj1131c	*gne*	UDP-glucose 4-epimerase	M	−1.85
Cj1133	*waaC*	Lipopolysaccharide core heptosyltransferase I	M	−1.94
Cj1134	*htrB*	Lipid A biosynthesis lauroyl acyltransferase	M	2.41
Cj1136		Glycosyltransferase	M	−29.66
Cj1137c		Hypothetical protein	M	22.54
Cj1138		Glycosyltransferase	M	2.66
Cj1139c	*wlaN*	β-1,3 galactosyltransferase	S	29.18
Cj1140	*cstIII*	α-2,3 sialyltransferase	G	25.08
Cj1141	*neuB1*	N-acetylneuraminate synthase	M	−1.85
Cj1142	*neuC1*	UDP-N-acetylglucosamine 2-epimerase	M	−2.85
Cj1146c	*waaV*	Glucosyltransferase	S	24.23
Cj1148	*waaF*	ADP-heptose--lipooligosaccharide heptosyltransferase II	M	−27.57
Cj1151c	*hldD*	ADP-L-glycero-D-manno-heptose-6-epimerase	M	−1.73
Cj1152c	*gmhB*	D-glycero-α-D-manno-heptose-1,7-bisphosphate 7-phosphatase	E	−24.54
Cj1221	*groEL*	Heat shock protein 60 kDa family chaperone	O	−1.32
Cj1228c	*htrA*	Serine protease	M	−1.12
Cj1239	*pdxA*	4-hydroxythreonine-4-phosphate dehydrogenase	H	−3.27
Cj1242	*ciaC*	Invasion antigen	-	−23.59
Cj1261	*racR*	Two-component system response regulator	K	3.63
Cj1279c		Putative fibronectin domain-containing lipoprotein	S	1.27
Cj1293	*pseB*	UDP-N-acetylglucosamine 4,6-dehydratase (inverting)	M	−2.76
Cj1294	*pseC*	UDP-4-amino-4,6-dideoxy-N-acetyl-β-L-altrosamine transaminase	E	−2.85
Cj1311	*pseF*	Pseudaminic acid cytidylyltransferase	M	2.42
Cj1312	*pseG*	UDP-2,4-diacetamido-2,4,6-trideoxy-β-L-altropyranose hydrolase	M	−27.16
Cj1313	*pseH*	Acetyltransferase	J	35.25
Cj1316c	*pseA*	Flagellin modification protein	D	−2.06
Cj1332	*ptmA*	Oxidoreductase (Flagellin modification)	N	−2.35
Cj1334	*maf3*	Motility accessory factor	S	−2.52
Cj1337	*pseE*	Motility accessory factor	S	−1.14
Cj1374c		dITP/XTP pyrophosphatase	F	3.84
Cj1408	*fliL*	Flagellar basal body-associated protein	N	−3.32
Cj1413c	*kpsS*	Capsular polysaccharide export system protein	M	−26.38
Cj1414c	*kpsC*	Capsular polysaccharide export system protein	M	2.53
Cj1415c	*cysC*	Cytidine diphosphoramidate kinase	P	−29.06
Cj1416c		CTP:phosphoglutamine cytidylyltransferase	M	−1.20
Cj1417c		Gamma-glutamyl-CDP-amidate hydrolase	S	−23.42
Cj1419c		Methyltransferase	Q	4.62
Cj1420c		Methyltransferase	Q	−1.63
Cj1421c		Possible sugar transferase	S	−2.48
Cj1422c		Hypothetical protein	S	−5.72
Cj1423c	*hddC*	D-glycero-α-D-manno-heptose 1-phosphate guanylyltransferase	JM	25.59
Cj1425c	*hddA*	D-glycero-alpha-D-manno-heptose 7-phosphate kinase	S	1.31
Cj1426c		Hypothetical protein	J	−2.00
Cj1429c		Hypothetical protein	S	23.07
Cj1430c	*rfbC*	dTDP-4-dehydrorhamnose 3,5-epimerase	M	−2.74
Cj1431c	*hddC*	Hypothetical protein	S	−1.81
Cj1433c		Hypothetical protein	H	−3.57
Cj1435c		Hypothetical protein	E	−1.77
Cj1436c		Hypothetical protein	E	−1.14
Cj1437c		Aminotransferase	E	−3.58
Cj1438c		Putative sugar transferase	S	−1.39
Cj1439c	*glf*	UDP-galactopyranose mutase	M	1.87
Cj1441c	*kfiD*	UDP-glucose 6-dehydrogenase	C	−2.80
Cj1442c		Sugar transferase	M	−2.63
Cj1443c	*kpsF*	D-arabinose-5-phosphate isomerase	M	−2.90
Cj1447c	*kpsT*	Capsule polysaccharide export ATP-binding protein	GM	3.58
Cj1462	*flgI*	Flagellar P-ring protein	N	2.20
Cj1463	*flgJ*	Hypothetical protein	MNO	1.80
Cj1478c	*cadF*	Outer membrane fibronectin-binding protein	M	−1.62
Cj1564		Methyl-accepting chemotaxis signal transduction protein	NT	−1.38
Cj1565c	*pflA*	Paralyzed flagella protein	N	−2.01
Cj1631c		DUF773 domain-containing protein	-	3.36

**Table 3 foods-13-01828-t003:** Differentially expressed proteins associated with iron uptake identified from *C. jejuni* strain 11168 biofilms in chicken juice. COG: Cluster of Orthologous Groups. **E**, amino acid transport and metabolism; **H**, coenzyme transport and metabolism; **M**, cell wall/membrane/envelope biogenesis; **P**, inorganic ion transport and metabolism; **S**, function unknown; **U**, intracellular trafficking, secretion, and vesicular transport; **V**, defense mechanisms.

Locus Tag	Genes	Description	COG	Log_2_ (Fold Ratio)
Cj0173c	*cfbpC*	Putative iron-uptake ABC transport system ATP-binding protein	E	−28.05
Cj0174c	*cfbpB*	Ferric iron ABC transporter, permease protein	P	−27.34
Cj0177		Hypothetical protein	S	−28.21
Cj0178		Putative outer membrane siderophore receptor	P	4.89
Cj0180	*exbD1*	Ferric siderophore transport system, biopolymer transport protein	U	24.51
Cj0181	*tonB1*	putative TonB-dependent receptor	M	3.56
Cj0755	*cfrA*	Ferric enterobactin uptake receptor	P	1.48
Cj1353	*ceuC*	Hypothetical protein	P	2.58
Cj1354	*ceuD*	Hypothetical protein	P	−26.47
Cj1355	*ceuE*	Enterochelin uptake periplasmic binding protein	P	−1.17
Cj1398	*feoB*	Ferrous iron transporter FeoB	P	−2.84
Cj1614	*chuA*	Haemin uptake system outer membrane receptor	P	−26.07
Cj1615	*chuB*	Haemin uptake system permease protein	P	−27.44
Cj1616	*chuC*	Haemin uptake system ATP-binding protein	HP	−9.31
Cj1630	*tonB2*	putative TonB-dependent receptor	U	−1.39
Cj1658		Iron permease	P	−1.05
Cj1661		Fe^2+^ ABC transporter, permease protein 1	V	−23.75
Cj1663		Fe^2+^ ABC transporter, ATP-binding subunit	V	−1.24

## Data Availability

The original contributions presented in the study are included in the article/supplementary material, further inquiries can be directed to the corresponding author.

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
