# Peer review of "Chicken Juice Enhances C. jejuni NCTC 11168 Biofilm Formation with Distinct Morphological Features and Altered Protein Expression"

_foods, 2024, doi:10.3390/foods13121828_

Round 1

Reviewer 1 Report

Comments and Suggestions for Authors

This research aims to understand the impact of chicken juice (CJ) on biofilm production of Campylobacter jejuni. The findings may be valuable to development of more effective control strategies for combatting this pathogen. However, the results presentation and data analysis in this paper are not detailed enough. Some problems listed as follows.

1. This paper measured biofilm using the CV method, however, an appropriate reference is missing.

2. Authors should explain why they prepared planktonic cells cultured for 16 hours and biofilms cells for 72 hours in their proteomic analysis experiments.

3. Figure 2 should be supplemented with the results of the statistical test and analysis, using lowercase letters or * symbols.

4. The length data of the scale bar should be labeled on the SEM or TEM result photographs, for example, presented similar as shown in Figure 5.

5. It is recommended that the font and font size of the vertical and horizontal coordinates in the all plots be consistent as much as possible.

6. The font size of the "sphere" in Figure 11 is too small to be read clearly, so it is suggested that it be modified.

7. The gene symbols in Tables 1 and 2 should be written in italicized font.

8. The discussion section of this paper seems too lengthy and needs to condense the most important elements.

9. The format of the references and their citation in the text of this paper do not meet the Journals requirements of Foods.

Author Response

Response to Reviewer 1 Comments

 Dear Reviewer:

Thank you for your thorough review and insightful comments. We appreciate your time and effort in helping us improve the manuscript. We have addressed each of your points below. We believe these revisions address your concerns and improve the overall quality of the manuscript.

  1. This paper measured biofilm using the CV method, however, an appropriate reference is missing.

Response: Thank you for pointing out the missing reference. We have added the following reference: O'Toole GA (2011) Microtiter dish biofilm formation assay. J Vis Exp (847): 2437.

  1. Authors should explain why they prepared planktonic cells cultured for 16 hours and biofilms cells for 72 hours in their proteomic analysis experiments.

Response: Thank you for your question about the culturing times. In the revised manuscript, we have explained our rationale for culturing planktonic cells for 16 hours and biofilm cells for 72 hours in the proteomic analysis experiments. “Planktonic cells cultured for 16 hours were chosen since this period is critical as it allows us to observe the cellular responses and protein expression profiles during a time when the cells are actively growing and metabolizing nutrients. On the other hand, biofilm cells were cultured for 72 hours to ensure the formation of a mature biofilm. A 72-hour culture period is sufficient to provide a comprehensive view of the proteomic changes associated with biofilm maturation and stability.”

  1. Figure 2 should be supplemented with the results of the statistical test and analysis, using lowercase letters or “*” symbols.

Response: Thank you for your suggestion regarding the statistical analysis. We have conducted the requested statistical test and replaced the graph with one using "*" symbols to indicate significance. The statistical method details are now included in the Methods section.

  1. The length data of the scale bar should be labeled on the SEM or TEM result photographs, for example, presented similar as shown in Figure 5.

Response: Thank you for noticing the missing information. We have labeled the length of the scale bar in the SEM images.

  1. It is recommended that the font and font size of the vertical and horizontal coordinates in the all plots be consistent as much as possible.

Response: Thank you for your attention to detail. We have standardized the font and font size of vertical and horizontal coordinates in all plots.

  1. The font size of the "sphere" in Figure 11 is too small to be read clearly, so it is suggested that it be modified.

Response: Thank you for pointing out the small font size. We increased the font size of the "sphere" label.  We are aware that the figure layout might have changed slightly due to database updates.

  1. The gene symbols in Tables 1 and 2 should be written in italicized font.

Response: Thank you for catching this formatting issue. Gene symbols in Tables 1, 2, and 3 are now italicized.

  1. The discussion section of this paper seems too lengthy and needs to condense the most important elements.

Response: Thank you for your suggestion on the discussion section. We have condensed the discussion section to focus on the most important findings.

  1. The format of the references and their citation in the text of this paper do not meet the Journal’s requirements of Foods.

Response: Thank you for clarifying the reference formatting requirements. Foods instructed that “Your references may be in any style, provided that you use consistent formatting throughout. It is essential to include author(s) name(s), journal or book title, article or chapter title (where required), year of publication, volume and issue (where appropriate), and pagination.” Therefore, we ensured consistent formatting throughout and haven't changed the specific style.

Reviewer 2 Report

Comments and Suggestions for Authors

Dear authors, the study is very interesting. I would like to make a few comments to contribute to the quality of the paper. 

C. jejuni NCTC 11168 strain was the only strain used in the study and should be referred to in full in all topics:

2.2. Measuring the growth line 103. 

2.3 Measurement of biofilm formation line 115

Also, in topic 2.2. Growth Measurement - Lines 90-105, it was described that the exudate was sterilised using a 0.2 μm Rapid-Flow sterile disposable polyethersulphone filter (Thermo Fisher Scientific), centrifuged at 20,817 x g 96 for 1 hour at 4oC and the resulting CJ was collected. Next, it was mentioned that bacterial contamination was confirmed by inoculating 100 μl of CJ onto MH agar supplemented with 5% lysed sheep's blood, which was incubated at 37oC for 48 hours. I don't understand what this contamination is... 

All legends for figures, graphs and tables should be revised. The captions should be self-explanatory, without the reader having to refer to the text to understand the issue.  

Has a control study been carried out under conditions other than those of CJ? It would be interesting to look at the difference in the protein profile between the planktonic cells and the biofilm.

Was a comparison made for BHI or MH culture media other than the growth curve? If so, you could include the results in the supplementary material and mention that this was done as well, but that only CJ was used for the subsequent analyses.

Author Response

Response to Reviewer 2 Comments

 Dear Reviewer:

Thank you for your careful review of our manuscript. We appreciate your insightful comments, which have helped us improve the clarity and completeness of our work. We believe that these revisions address your concerns and improve the clarity of our work.  We welcome your further feedback.

 jejuni NCTC 11168 strain was the only strain used in the study and should be referred to in full in all topics:

2.2. Measuring the growth line 103.

2.3 Measurement of biofilm formation line 115

Response: We have addressed your suggestion and ensured that the full strain name, C. jejuni NCTC 11168, is used in line 103 and 115.

Also, in topic 2.2. Growth Measurement - Lines 90-105, it was described that the exudate was sterilised using a 0.2 μm Rapid-Flow sterile disposable polyethersulphone filter (Thermo Fisher Scientific), centrifuged at 20,817 x g 96 for 1 hour at 4oC and the resulting CJ was collected. Next, it was mentioned that bacterial contamination was confirmed by inoculating 100 μl of CJ onto MH agar supplemented with 5% lysed sheep's blood, which was incubated at 37oC for 48 hours. I don't understand what this contamination is...

Response: A follow-up culture testing was performed to double-check sterility.

All legends for figures, graphs and tables should be revised. The captions should be self-explanatory, without the reader having to refer to the text to understand the issue. 

Response: Thank you for pointing this out. We have carefully reviewed all figure, graph, and table legends and revised them to be more self-explanatory. The revised captions will provide a clear understanding of the data presented without requiring reference to the main text.

Has a control study been carried out under conditions other than those of CJ? It would be interesting to look at the difference in the protein profile between the planktonic cells and the biofilm.

Response: We appreciate your suggestion for a control study using a different growth condition. We acknowledge that comparing the proteome profiles of planktonic cells in CJ to those in a different broth medium (e.g., BHI or MH) could be informative. However, due to cost constraints, we were only able to perform the proteomic analysis on cells grown in CJ.

The protein profile presented here compares the proteome of C. jejuni biofilms grown in CJ with that of planktonic cells, also grown in CJ, as a control (refer to Methods for details).

Was a comparison made for BHI or MH culture media other than the growth curve? If so, you could include the results in the supplementary material and mention that this was done as well, but that only CJ was used for the subsequent analyses.

Response: As you mentioned, we did compare BHI and MH media for growth curve and biofilm formation assays. However, due to the cost constraints associated with proteomic analysis, we limited this analysis to CJ for protein profile investigation.  We have added a sentence in the manuscript to acknowledge this limitation and mention that BHI and MH media were only compared for growth and biofilm formation.

Reviewer 3 Report

Comments and Suggestions for Authors

Epidemiological data from recent years show steadily increasing rates of campylobacteriosis cases, with food as a source (including poultry meat). Thus, it seems reasonable for the authors to undertake a research issue aimed at assessing the ability of Campylobacter jejuni to form a biofilm and alter protein expression. Below are some comments to the authors:

Title - give full genre name and put in italics

Abstract:

- write the species name in italics

Introduction:

- line 30 - please provide the trends of the disease in recent years, are there decreases or increases in the disease ? Which continents are struggling with more cases of campylobacteriosis ?

- line 32 - please add information on the patient groups most likely to be infected with C. jejuni

Material and methods:

- line 83 - Please elaborate on the abbreviation for NCTC

- line 104 - please complete the name of the spectrophotometer

Results:

- fig. 1 i 2 - in the description of the figure, please explain all abbreviations that appear in the legend

- line 223 - has statistical analysis been done for this data ? Please include in the methodology

- line 227 - duplicate ? the methodology states that the experiments were performed in triplicate 

- for figures 3 - 5, please describe exactly what is seen in point A, B, etc.

- tables 1, 2, 3 - Please put the names of the genes in italics, explain the abbreviation for COG under the table

Author Response

Response to Reviewer 3 Comments

 Dear Reviewer:

We appreciate your time and effort in helping us improve the manuscript. We have addressed each of your points below. We believe these revisions address your concerns and improve the overall quality of the manuscript. We believe these revisions address your concerns and improve the clarity and completeness of our manuscript.  We welcome your further feedback.

 Reviewer 3

Epidemiological data from recent years show steadily increasing rates of campylobacteriosis cases, with food as a source (including poultry meat). Thus, it seems reasonable for the authors to undertake a research issue aimed at assessing the ability of Campylobacter jejuni to form a biofilm and alter protein expression. Below are some comments to the authors:

Title - give full genre name and put in italics

Response: Thank you for pointing that out. We have revised the full genus name, C. jejuni NCTC 11168, and put it in italics.

Abstract:

- write the species name in italics

Response: Thank you for comment. We have italicized the species name.

Introduction:

- line 30 - please provide the trends of the disease in recent years, are there decreases or increases in the disease? Which continents are struggling with more cases of campylobacteriosis?

Response: Thank you for your suggestion. We have added a sentence about the recent trends in the disease: “In 2020, the average reported incidence of Campylobacter infection declined in the United States and most European countries compared to 2014-2019 (2). However, Czech Republic had the highest worldwide incidence that year (215 per 100,000 popu-lation). This was followed by Australia (146.8 per 100,000 in 2016) and New Zealand (126.1 per 100,000 in 2019).

- line 32 - please add information on the patient groups most likely to be infected with C. jejuni

Response: Thank you for your comment. We have added information on patient groups susceptible to C. jejuni infection: “Populations most susceptible to C. jejuni infection include young children, elderly individuals, and immunocompromised patients.

Material and methods:

- line 83 - Please elaborate on the abbreviation for NCTC

Response: Thank you for your suggestion. We have elaborated on the abbreviation for NCTC (National Collection of Type Cultures).

- line 104 - please complete the name of the spectrophotometer

Response: Thank you for noticing this. We have added the complete name of the spectrophotometer: SmartSpec Plus UV/VIS Spectrophotometer (Bio-Rad, Hercules, CA).

Results:

- fig. 1 i 2 - in the description of the figure, please explain all abbreviations that appear in the legend

Response: Thank you for your comment. We have clarified all abbreviations used in the figure legends.

- line 223 - has statistical analysis been done for this data? Please include in the methodology

Response: Thank you for this suggestion. We have added a sentence to the Methodology section explaining the statistical analysis: “To assess growth kinetics, the standard deviation was calculated for the mean growth kinetic values of each condition.” This information is also reflected in the error bars of the growth kinetics graph.

- line 227 - duplicate? the methodology states that the experiments were performed in triplicate

Response: Thank you for catching this error. We have corrected the typographical error. The text now reads "performed in triplicate" to reflect the methodology.

- for figures 3 - 5, please describe exactly what is seen in point A, B, etc.

Response: Thank you for your suggestion. We have added detailed descriptions of the observations at points A, B, etc. in lines 263-273.

- tables 1, 2, 3 - Please put the names of the genes in italics, explain the abbreviation for COG under the table

Response: Thank you for pointing this out. We have italicized the gene names and added an explanation of the abbreviation COG under the respective tables.

Round 2

Reviewer 1 Report

Comments and Suggestions for Authors

The authors adequately responded to the comments of the reviewers and the revision is satisfactory